# Discrete Element Method Modelling of the Diametral Compression of Starch Agglomerates

**DOI:** 10.3390/ma13040932

**Published:** 2020-02-20

**Authors:** Józef Horabik, Joanna Wiącek, Piotr Parafiniuk, Mateusz Stasiak, Maciej Bańda, Rafał Kobyłka, Marek Molenda

**Affiliations:** Institute of Agrophysics, Polish Academy of Sciences, Doświadczalna 4, 20-290 Lublin, Poland; j.wiacek@ipan.lublin.pl (J.W.); p.parafiniuk@ipan.lublin.pl (P.P.); m.stasiak@ipan.lublin.pl (M.S.); m.banda@ipan.lublin.pl (M.B.); r.kobylka@ipan.lublin.pl (R.K.); m.molenda@ipan.lublin.pl (M.M.)

**Keywords:** agglomerates, diametral-compression test, tensile strength, bonded-particle model

## Abstract

Starch agglomerates are widely applied in the pharmaceutical, agricultural, and food industries. The formation of potato starch tablets and their diametral compression were simulated numerically and verified in a laboratory experiment to analyse the microscopic mechanisms of the compaction and the origins of their breakage strength. Discrete element method (DEM) simulations were performed using EDEM software. Samples comprised of 120,000 spherical particles with radii normally distributed in the range of 5–36 μm were compacted in a cylindrical die with a diameter of 2.5 cm. The linear elastic–plastic constitutive contact model with a parallel bonded-particle model (BPM) was used to model the diametral compression. DEM simulations indicated that the BPM, together with the linear elastic–plastic contact model, could describe the brittle, semi-brittle, or ductile breakage mode, depending on the ratio of the strength to Young’s modulus of the bond and the bond-to-contact elasticity ratio. Experiments confirmed the findings of the DEM simulations and indicated that potato starch (PS) agglomerates can behave as a brittle, semi-brittle, or ductile material, depending on the applied binder. The PS agglomerates without any additives behaved as a semi-brittle material. The addition of 5% of ground sugar resulted in the brittle breakage mode. The addition of 5% gluten resulted in the ductile breakage mode.

## 1. Introduction

Agglomeration of powders is one of the unit operations for improving the characteristics and functionality of the final product. It is widely applied in the pharmaceutical, agricultural, food, mineral, metallurgy, and fuel biomass industries [1]. The quality of compacts is influenced by the powder and binder properties, as well as the compaction pressure [2]. The strength of the agglomerates is one of the most important properties affecting the final product [3]. Two of the most frequently applied tests for assessing the tensile strength of compacts are the diametral compression test and the uniaxial, unconfined compression test.

During mechanical tests, agglomerates are subjected to external loads, resulting in elastic and plastic deformation and ultimate failure. After the peak value of the stress is reached, fracture of the agglomerate occurs, and the stress decreases to a residual stress state [4]. Depending on the amount of plastic deformation needed to cause failure and shape of the force–displacement response, agglomerates can be classified as brittle, semi-brittle, or ductile. Strongly brittle materials, such as rock, glass, and monolithic ceramics, exhibit no detectable plastic strain prior to failure and a sharp peak of the force–displacement response [5,6]. Brittle mechanical behaviour involves localised cracks [7]. Materials with intermediate compacted structures and an intermediate bond stiffness characterised by a round force–displacement response are considered semi-brittle. Semi-brittle materials include polymers [8,9], inorganic materials connected by a soft binder [10], and pharmaceutical granules [11]. Materials with loose structures and a low bond stiffness that exhibit a significant amount of fully plastic deformation prior to failure are considered ductile [5]. These include sand–asphalt mixtures [12]; soft, high-porosity materials, such as aerogel composites [13]; pellets of biomass [14]; and lightweight food products of the extrusion process [15]. Food and pharmaceutical granules (and tablets) can exhibit brittle [16], semi-brittle, or ductile [17] behaviour, depending on the material composition, applied binder, and porosity of the agglomerates. Dhanalakshmi and Bhattacharya [18] demonstrated that pressure-agglomerated corn starch exhibited brittle, semi-brittle, and ductile behaviour when glued by pre-gelatinised starch, sugar, and vegetable oil, respectively. This is consistent with findings indicating that the properties of adhesives (ductile or brittle) significantly affect the strength behaviour of joints [19].

Theories regarding failure mechanisms have been developed, but the behaviour of some agglomerates remains unclear [7]. The formation mechanisms of adhesion bonds during agglomeration, the fracture process of compacts during mechanical tests, and the conditions of the transition among localised brittle, semi-brittle, and ductile deformation with the change of the material parameters require deeper understanding [2,20,21].

The diametral compression test (also known as the Brazilian disk test) is widely used for determining the tensile strength of circular objects, owing to its simplicity (Figure 1b). The stress component *σ_y_* is compressive, while the stress component *σ_x_* (in the direction perpendicular to the load) is tensile in locations close to the disc centre and compressive at the upper most locations close to the loading plates. For *x* = 0 and *y* = 0, the stresses *σ_x_* and *σ_y_* are the principal stresses *σ*_1_ and *σ*_2_, respectively [22]. The tensile strength *σ_f_* of the agglomerates is identified by the maximum tensile stress *σ*_1,max_ in the direction perpendicular to the load in the centre of the disc (*x* = 0, *y* = 0) [23]:(1)$$\sigma_{f}=\sigma_{1,\max}^{}=\frac{P_{f}}{\pi Rt},$$
where $P_{f}$ is the failure load, *R* is the radius of the agglomerate, and *t* is its thickness.

The mechanical properties of compacts can be related to the properties of single granules using the micromechanical approach. The discrete element method (DEM) proposed by Cundall and Strack [24] is a very promising tool for obtaining insight into the problem of powder compaction at the particle level. To simulate the internal structure of the agglomerate and its breakage, the method has been extended with the bonded-particle model (BPM), as proposed by Potyondy and Cundall [25]. The brittle and semi-brittle breakage processes can be approximated well using the BPM. Three regions of the proportion of the elastic modulus of bond *E^b^* to the elastic modulus of particle *E* can be distinguished: *E^b^* = *E* [25,26,27], *E^b^* > *E* [28], and *E^b^* < *E* [3,29]. For *E^b^* ≥ *E*, the DEM models provide typical brittle behaviour, characterised by a rapid decrease of the force–displacement response [27]. As the *E^b^*/*E* ratio decreases, the breakage mode evolves from brittle to semi-brittle, with a round force–displacement response [30,31,32].

The objective of this study was to investigate the possibility of numerically reproducing the brittle, semi-brittle, and ductile breakage modes of agglomerates by using the DEM together with a parallel BPM.

## 2. Materials and Methods 

### 2.1. Materials

Potato starch (PS) produced by Melvit S.A., Warsaw, Poland and ground sugar (GS) were purchased from a local supermarket. Wheat gluten (WG) was obtained from Sigma–Aldrich Ltd., Poznań, Poland.

### 2.2. Methods

The initial moisture content of the PS was 10%. The PS without any additives or with the addition of 5% gluten or sugar was moistened for 48 h in a humidifier in a closed space at a relative humidity of 70% and a temperature of 21 °C, in order to increase the moisture content to 17%. Steam conditioning was applied for 60 s prior to agglomeration to increase the solubility of the additives, i.e., sugar or gluten. To minimise the PS gelatinisation, the temperature of the steam was 60 °C [33]. The steaming increased the moisture content by no more than 0.2%. The moisture content (wet basis) was measured gravimetrically by weighting a 10 g sample in the wet state and after drying at 105 °C for 24 h.

Samples of 0.5 g of PS were compacted in a stainless-steel cylindrical die with a diameter of 10 mm, at a displacement rate of 0.02 mm s^−1^ up to compaction pressures of *σ_z_* = 38, 76, 114, and 153 MPa (Figure 1a). The height of the tablet after unloading and relaxation, determined by a caliper with an accuracy of 0.01 mm, was 4.8 ± 0.1 mm. The tensile strength was determined via a diametral compression test with a displacement rate of 0.033 mm s^−1^. The compression tests were performed immediately after the compaction process to avoid the effect of storage time on the strength. The reference material was PS with a moisture content of 17%, without additives. All variants of the experiments were repeated 10 times.

## 3. Discrete Element Method Setup

The linear hysteretic spring contact model introduced by Walton and Brown [34] was used for simulations. The model was equipped with linear adhesion according to concept of Luding [35]. The contact force–displacement scheme in the normal direction considered the plastic contact deformation, elastic unloading and reloading, and attractive adhesion forces:(2)$$f_{n}=\{\begin{array}{lll} k_{1}\delta_{n} & loading & k_{2}(\delta_{n}-\delta_{n,0})\geq k_{1}\delta_{n}\\ k_{2}(\delta_{n}-\delta_{n,0}) & unloading/reloading & k_{1}\delta_{n}>k_{2}(\delta_{n}-\delta_{n,0})>-k_{c}\delta_{n}\\ -k_{c}\delta_{n} & unloading & -k_{c}\delta_{n}\geq k_{2}(\delta_{n}-\delta_{n,0}) \end{array},$$
where *f_n_* is the contact force, *k*_1_ is the loading (plastic) stiffness, *k*_2_ is the unloading (elastic) stiffness, *k_c_* is the adhesive stiffness, and *δ_n_* is the overlap in the normal direction (Figure 2a). During unloading, the force *f_n_* decreased to zero at the overlap *δ_n,_*_0_. The plastic stiffness *k*_1_ was related to the yield strength *p_y_* of a particle as follows [36,37]:(3)$$p_{y}=\frac{2E^{}}{\pi }\sqrt{\frac{\delta_{n,y}}{r}},$$
(4)$$k_{1}=5r^{*}\min(p_{y,i},p_{y,j}),$$
where *E* is Young’s modulus, *r* is the radius of a particle, *δ_n,y_* is the yielding overlap, $r^{*}=r_{i}r_{j}/(r_{i}+r_{j})$ is the equivalent radius of the contacting particles, and *p_y,i_* and *p_y,j_* are the yield strengths of particles *i* and *j*, respectively. The energy dissipation in the normal direction was due to the difference between the loading and unloading stiffness and the elastic stiffness *k*_2_. For unloading and reloading, *k*_2_ was related to *k*_1_ through the restitution coefficient *e* [37], as follows:(5)$$e=\sqrt{\frac{k_{1}}{k_{2}}}.$$

The particle–particle force in the tangential direction *f_t_* was updated incrementally as follows:(6)$$f_{t}^{}=\{\begin{array}{lll} {\left(f_{t}^{}\right)}_{0}+k_{t}\Delta \delta_{t}+f_{t}^{d}, & if & \left|f_{t}^{}\right|<\mu_{p-p}f_{n}^{}\\ \frac{\Delta \delta_{t}}{\left|\Delta \delta_{t}\right|}\mu_{p-p}f_{n}^{}, & if & \left|f_{t}^{}\right|\geq \mu_{p-p}f_{n}^{} \end{array},$$
where (*f_t_*)_0_ is the tangential force at the end of the previous timestep; *k_t_* and *δ_t_* are the stiffness and overlap in the tangential direction, respectively; and *μ_p-p_* is the particle–particle friction coefficient. The stiffness in the tangential direction *k_t_* was assumed to be equal to the stiffness in the normal direction. The velocity-dependent dissipative component $f_{t}^{d}$ of the tangential force *f_t_* is defined as follows [37,38]:(7)$$f_{d}^{t}=-\sqrt{\frac{4m^{*}k_{t}}{1+{\left(\frac{\pi }{\ln e}\right)}^{2}}}v_{t},$$
where $m^{*}=m_{i}m_{j}/(m_{i}+m_{j})$ is the equivalent mass of the contacting particles, and *v_t_* is the relative velocity in the tangential direction.

The adhesion model relates the force to the contact area as follows: $f_{n}=-k\pi r^{*}\delta_{n}$, where *k* is the adhesion energy density [37]. Hence, the adhesive stiffness *k_c_* has a linear relationship with the adhesion energy density *k* and the equivalent radius of the contacting particles *r^*^*:(8)$$k_{c}=\pi kr^{*}.$$

The BPM model proposed by Potyondy and Cundall [25] was introduced to describe the interparticle bonding forces and moments (Figure 2b). The forces and moments of the bond connections between two particles were calculated incrementally [37]:(9)$$\begin{array}{l} \Delta f_{n}^{b}=-v_{n}k_{n}^{b}A\Delta t\\ \Delta f_{t}^{b}=-v_{t}k_{t}^{b}A\Delta t\\ \Delta M_{n}^{b}=-\omega_{n}k_{t}^{b}J\Delta t\\ \Delta M_{t}^{b}=-\omega_{t}k_{n}^{b}\frac{J}{2}\Delta t \end{array}$$
where *v_n_* and *v_t_* are the relative velocities in the normal and tangential directions, respectively; $k_{n}^{b}$ and $k_{t}^{b}$ are the stiffnesses in the normal and tangential directions, respectively; $A=\pi r_{b}^{2}$ and $J=\frac{\pi r_{b}^{4}}{2}$ are the area and moment of inertia of the bond cross section, respectively; *r_b_* is the radius of the bond; and Δ*t* is the time increment.

The Young’s modulus of the bond *E^b^* is:(10)$$E^{b}=k_{n}^{b}(r_{i}+r_{j}),$$
where $k_{n}^{b}$ is the stiffness in the normal direction, *r_i_* and *r_j_* are radii of the contacting particles *i* and *j*, respectively.

The bond is broken when the maximum normal stress $\sigma_{\max}^{b}$ exceeds the tension strength *σ^c^* or the maximum tangent stress $\tau_{\max}^{b}$ exceeds the shear strength *τ^c^*:(11)$$\begin{array}{l} \sigma_{\max}^{b}=\frac{-f_{n}^{b}}{A}+\frac{2M_{t}^{b}}{J}r_{b}>\sigma^{c}\\ \tau_{\max}^{b}=\frac{-f_{t}^{b}}{A}+\frac{M_{n}^{b}}{J}r_{b}>\tau^{c}. \end{array}$$

The DEM simulation was separated into five stages: filling, compaction, unloading, relaxation, and diametral compression, as shown in Figure 3. The cylindrical die had a diameter of 2.5 mm and a height of 6 mm, yielding a DEM/experiment scale ratio of 1:4 for the agglomerate. Spherical particles simulating PS [39] with radii normally distributed (mean value of 20 μm, standard deviation of 7.5 μm, range of 5–36 μm) (Table 1) were generated randomly in the die (Figure 3a). The assembly consisted of 120,000 particles. After settlement, the particles were compacted with a piston axial velocity of 10^−3^ m s^−1^ to the assumed value of the agglomeration pressure $\sigma_{z}^{c}$ (Figure 3b). After the desired agglomeration pressure was reached, the sample was unloaded with the same velocity (Figure 3c). After complete unloading, the mould was removed, and relaxation of the agglomerate was performed (Figure 3d). Simulations of the diametral compression were performed, with the displacement velocity in the range from 10^−5^ to 10^−3^ m s^−1^.

For filling and compaction, the linear hysteretic model was applied. In accordance with the findings of He et al. [40], the plastic stiffness *k*_1_ of 3 × 10^4^ N m^−1^ was adjusted to provide an in-die agglomerate porosity of *ε* < 0.1 under the compaction pressure of 153 MPa. For the assumed Young’s modulus of the PS (2.5 × 10^3^ MPa) [41], the corresponding value of the yielding overlap *δ_y_* (Equation (3)) was 0.0354*r*. The adopted value of the restitution coefficient (*e* = 0.5), which is typical for biological materials [42], provided an elastic stiffness coefficient of *k*_2_ = 1.2 × 10^5^ N m^−1^.

The linear hysteretic model was extended to account for linear adhesion, in order to minimize changes in the structure of the agglomerate during unloading and relaxation. The adhesive stiffness *k_c_* was set as 300 N m^−1^ [43] to match the experimental tensile strength of the agglomerate. For the diametral compression test, the adhesion model was replaced with the parallel BPM to prevent simulations from appearing on secondary adhesive contacts, which might affect the resultant tensile strength of the agglomerates.

The Poisson ratio *ν* = 0.25 and the coefficient of restitution *e* = 0.5 were adopted for the DEM simulations, as is typical for materials of biological origin [42]. The particle–wall friction coefficient *μ_p-w_* of 0.1 was taken as half of the coefficient of the sliding friction of PS against stainless steel [44], in order to account for lubrication with magnesium stearate [45]. The coefficient of particle–particle friction *μ_p-p_* was set to 0.5 to reproduce the typical values of 30°−35° for the angle of internal friction of PS [46]. The coefficient of rolling friction *m_r_* was set as 0.01, in accordance with the best-fit values from similar studies on spherical particles [47].

To reduce the computational time, the particle density was increased by a factor of 10^6^. To keep the gravitational force unchanged, the gravitational acceleration was reduced by a factor of 10^6^. As shown in [43], scaling the density by a factor of 10^6^ did not change the shape of the *σ*_1_(Δ*L*/*D*) characteristics, and reduced the tensile strength *σ_f_* by 1.2% of the strength of the sample without density scaling. Time integration was performed with steps of 3 × 10^−6^ s, i.e., 14% of the Rayleigh timestep [48]. The EDEM software package [37] was used for the numerical simulations.

The components of the macroscopic “quasi-static” stress tensor *σ_ij_* in the system of particles averaged over all the contacts in volume *V* were determined as the dyadic product of the contact force $f_{j}^{c}$ vector at contact *c* and the branch vector $l_{i}^{c}$ connecting two contacting particles *a* and *b*, according to the concept presented by Christoffersen et al. [49]:(12)$$\begin{array}{cc} \sigma_{ij}=\frac{1}{V}\sum_{c=1}^{N_{c}}l_{i}^{c}f_{j}^{c} & \left(i,j=x,y,z\right) \end{array}.$$

The representative volume element (RVE) of the particle contacts used for the calculation of the stress components was a cuboid (0.25 × 0.25 × 1.2 mm) moved along the vertical direction to obtain the stress profile along the loading direction (Figure 3e) [43]. The average number of contacts in the RVE was 19,000.

## 4. Results

### 4.1. Tensile Strength of Agglomerates

To adjust the bond and the particle parameters to the best-fit experimental data of the compaction process and the strength test of the agglomerates, preliminary simulations and analyses of the effects of different parameters on the relationship between the tension stress *σ*_1_ and the diametral deformation Δ*L*/*D*, the tensile strength of the agglomerates *σ_f_* during diametral compression, and the breakage mode were performed. These simulations and analyses included the shear strength of the bond *τ^c^*, the tensile strength of the bond *σ^c^*, the Young’s modulus of the bond *E^b^*, and the residual overlap at the instant of bond initiation *δ_n_*_,0_.

#### 4.1.1. Impact of Deformation Velocity

To verify the impact of the deformation velocity on the tensile strength of the agglomerates, preliminary simulations of the diametral compression were performed for *σ*^c^ = 10 MPa and *E^b^* = 200 MPa, as well as the deformation velocity in the range from 10^−5^ to 10^−3^ m s^−1^. The increase of the velocity from 10^−5^ to 10^−4^ m s^−1^ did not change considerably the shape of the *σ*_1_(Δ*L*/*D*) characteristic or the tensile strength *σ_f_*. An increase of the velocity to 10^−3^ m s^−1^ increased the *σ_f_* by an amount of 4%, as compared with the lowest velocity case (Figure 4). Taking into account the computing time for the simulations, this level of error was considered as acceptable for the purpose of this study.

#### 4.1.2. Impact of the Ratio of Shear to Tensile Strength of the Bond

The relationship between the tension stress *σ*_1_ and the diametral deformation Δ*L*/*D* was not significantly influenced by the change in the *τ^c^*/*σ^c^* ratio (Figure 5a), while the tensile strength of the agglomerates *σ_f_* increased to a saturation value, stabilising for a *τ^c^*/*σ^c^* ratio slightly higher than 1 (Figure 5b). Analysis of the distribution of the bond normal and tangent forces performed for the *τ^c^*/*σ^c^* ratio of 1 indicated that the mean bond tangent force ${\bar{f}}_{t}^{b}$ at $\sigma_{1,\max}^{}$ ranged from 0.35 to 0.47 of the mean bond normal tension force ${\bar{f}}_{n}^{b}$ for the ductile and brittle breakage modes, respectively. This is consistent with the experimental findings of Jonsén et al. [23], that tension is the primary failure mode in the diametral compression test and is opposite to the shear primary failure mode in the uniaxial compression test, as reported by He et al. [4].

#### 4.1.3. Impact of Strength and Young’s Modulus of Bond

For a constant value of the bond of Young’s modulus, the tensile strength of the agglomerate *σ_f_* increased nonlinearly to a limiting value, indicating a qualitative change in the behaviour of the agglomerates with an increase in the tensile (and shear *σ^c^* = *τ^c^*) strength of the bond (Figure 6a). In the entire range of change of the *σ*_1_ and *σ_f_* with the *σ^c^* change, the following two regions were distinguished: (1) an almost linear increase of the *σ_f_*, with an accompanying round and a slowly flattening *σ*_1_(Δ*L*/*D*) relationship with a clear maximum (typical for semi-brittle breakage); and (2) the saturation of *σ_f_* with a further increase in *σ^c^*_,_ with an accompanying growing range of the deformation Δ*L*/*D* with a constant value of *σ*_1_ (typical for ductile breakage).

The change of Young’s modulus of the bond had an opposite effect on the qualitative change of the breakage behaviour of the agglomerates than did the bond strength (Figure 6b). With an increase in Young’s modulus of the bond, the breakage mode changed from ductile to semi-brittle at *E^b^* = 100 MPa. With a further increase in Young’s modulus, the *σ*_1_(Δ*L*/*L*) characteristics (typical for the semi-brittle breakage mode) evolved towards the brittle mode, i.e., exhibited a sudden drop of *σ*_1_ after approaching the peak stress. There was no clear threshold for the transition from semi-brittle to brittle behaviour.

#### 4.1.4. Impact of Ratio of Bond Strength to Young’s Modulus

Combining the effects of the Young’s modulus and the bond strength on the mechanical characteristics *σ*_1_(Δ*L*/*L*), the transition from ductile to semi-brittle breakage can be attributed to the *σ^c^*/*E^b^* ratio (Figure 7a). For *σ_z_* = 153 MPa, a ratio of *σ^c^*/*E^b^* < 0.1 provided semi-brittle breakage, whereas a ratio of *σ^c^*/*E^b^* > 0.15 provided ductile breakage (Figure 7b).

#### 4.1.5. Impact of Bond Cross-Sectional Area

A very clear transition between the semi-brittle and ductile behaviour was observed over the entire range of the compaction pressure. Figure 8a presents the *σ*_1_(Δ*L*/*L*) characteristics for selected values of the ratio of the mean overlap at the instant of bond initiation to the mean particle radius ${\bar{\delta }}_{n,0}/\bar{r}$. The mean overlap at the instant of bond initiation (${\bar{\delta }}_{n,0}$) did not increase linearly with the increase of the compaction pressure *σ_z_*; rather, the increase was slower. Thus, the bond cross-sectional area also exhibited a slower-than-linear increase with the increasing compaction pressure. The consequence of this is discussed in Section 4.2.1. The threshold value of the *σ^c^*/*E^b^* ratio for the semi-brittle and ductile transitions increased with the ${\bar{\delta }}_{n,0}/\bar{r}$ ratio (Figure 8b).

#### 4.1.6. Breakage Modes

Figure 9 presents examples of typical mechanical characteristics of the tension stress vs. the diametral deformation of PS tablets, as well as fitted DEM approximations illustrating brittle, semi-brittle, and ductile breakage modes. Individual PS tablets without any additives behaved as semi-brittle or brittle materials (Figure 9a). For a compaction pressure of 153 MPa, the tensile strength was 0.84 ± 0.1 MPa. The addition of sugar or gluten resulted in the semi-brittle breakage mode and reduced the breakage strength to 0.62 ± 0.03 and 0.47 ± 0.06 MPa for tablets with sugar and gluten, respectively. Steaming of the PS with the addition of sugar increased the tensile strength of the agglomerates to 1.1 ± 0.15 MPa and changed the breakage mode to brittle (Figure 9b). Steaming of the PS with the addition of gluten resulted in a two-fold decrease of the tensile strength (0.26 ± 0.06 MPa) and changed the breakage mode from semi-brittle to ductile. Steaming of the PS without any additives reduced the tensile strength of the agglomerates to 0.51 ± 0.06 MPa, and did not change the breakage mode.

The best-fitted DEM models approximating the analysed mechanical characteristics indicated considerable differences in the bond parameters between particular cases. For PS with no additives or steaming, the Young’s modulus of the bond was in the range of 200–300 MPa, and the tensile strength of the bond was in the range of 10–12 MPa. Steaming of the PS reduced the tensile strength of the bond to 7.5 MPa. Addition of sugar or gluten without steaming increased the Young’s modulus of the bond stiffness to 500 MPa and reduced the tensile strength of the bond to 8 MPa. The effect of steaming was opposite for PS with the addition of sugar and gluten. In the case of the addition of sugar, steaming slightly reduced the Young’s modulus of the bond (400 MPa) and increased the tensile strength from 8 to 18 MPa, owing to the crystallisation of the solubilised sugar after cooling. In the case of the addition of gluten, steaming resulted in a 20-fold reduction of the Young’s modulus of the bond (from 500 to 25 MPa), and did not change the tensile strength. This effect is attributed to the action of protein fibrils, which create elastic binding sites between PS particles [50].

The breakage mode influenced the structural changes of the tablets during diametral compression. For the PS tablets without any additives and the tablets with the addition of sugar or gluten without steaming, there was a single crack, almost aligned with the loading direction, or a Y-shaped crack, indicating the partition of the tension and shear interaction in the failure (PS, PS + WG, and PS + GS in Figure 10a). A single crack along the loading direction appeared for the steamed PS with the addition of sugar, i.e., the brittle breakage mode and tension crack. Steaming of the PS with the addition of gluten (ductile breakage) resulted in diffuse X-shaped cracks in locations close the tablet centre, as well as crushing in locations close to the loading platens.

The shapes of the breakage patterns of the tablets in the DEM simulations reproduced, to some degree, the shapes of the experimentally obtained patterns. The parameters with the largest influence were the compaction pressure *σ_z_* and the *σ^c^*/*E^b^* ratio (Figure 10b). With the increase of the agglomeration pressure and the decrease of the *σ^c^*/*E^b^* ratio, the X-shaped conjugate cracks changed to Y-shaped cracks for moderately compacted agglomerates (*σ_z_* = 76 MPa). A single crack aligned with the direction of the loading occurred for the highest level of the agglomeration pressure and the lowest value of the *σ^c^*/*E^b^* ratio. The gradual change of shape of the crack pattern with the changes of the compaction pressure and the *σ^c^*/*E^b^* ratio obtained in the DEM simulations are consistent with the results of the experimental study of the breakage of three-dimensional (3D)-printed agglomerates performed by Ge et al. [51]. Their study indicates that brittle breakage is typical for dense structures with a high bond stiffness, and that ductile breakage is typical for loose structures with a low bond stiffness.

There are some similarities and differences in particle–particle interactions and the bond interactions between the three indicated breakage modes. For all three breakage modes, the tension stress *σ*_1_ during loading was related almost linearly to the mean value of the magnitude of the contact normal force $\left|{\bar{f}}_{n}\right|$ (Figure 11a), as the result of the same value of the contact stiffness *k*_1_ and *k*_2_ applied for all simulations. Figure 11b presents profiles of the normalized average stress component *σ_x_*/*σ_x_*_,max_ along the *y* direction. Negative values correspond to the compressive stress, and positive values correspond to the tension stress. Stress profiles in the *y* direction were very similar for all three breakage modes: the stress was tensile in the central part of the disc and compressive in locations close to the loading platens. The shape of this stress profile was consistent with the findings of experimental [23], theoretical [22], finite element [52], and DEM [43] studies. These studies have indicated the domination of the tension stress in locations close to centre of the disc, which results in cracking, and domination of the compressive stress in locations close to the loading platens, which results in crushing.

The differences in the *σ*_1_(Δ*L*/*D*) relationships between the ductile, semi-brittle, and brittle breakage modes can be attributed to the difference in the dynamics of bond breakage when approaching the peak of *σ*_1_. Figure 11c illustrates the normalized rate of the breakage of bonds for three breakage modes. For the ductile breakage mode, the rate was almost constant during the entire process of deformation. For the brittle and semi-brittle breakage modes, the maximum of the rate was observed. The brittle and semi-brittle breakage modes were initiated at the instance of the highest increase in the rate of bond breakage. This indicates a very rapid change in the rate of bond breakage in the case of the brittle and semi-brittle breakage modes, and an almost constant rate in the case of the ductile breakage mode. Therefore, it seems that the course of change of the rate of breakage of bonds during deformation can be used to distinguish ductile and semi-brittle behaviour. The difference in the dynamic of breakage of bonds resulted in a difference in the rate of change of the kinetic energy of particles during deformation (Figure 11d). Kinetic energy started to increase rapidly at the instance of the *σ*_1_ peak, in the case of the brittle and semi-brittle breakage modes, and increased much more slowly but slightly faster than linearly during the entire process of deformation in the case of the ductile breakage mode.

There was no clear threshold for the transition from semi-brittle to brittle behaviour, but the transition was diffuse, owing to the gradual change in the partition of the intact bond force and the contact force in the total resistance to loading with the change in the bond-to-contact elasticity ratio (*E^b^*/*E*). In the bond-to-contact elasticity-ratio range of 0.12 to 0.2, the brittle and semi-brittle modes were observed alternately, owing to the effects of other materials and process parameters, such as the compaction pressure, additives, and steaming, which modified the elasticity, strength, and compaction of the tablets (Figure 10). For *E^b^* ≥ *E*, i.e., when the dominating interaction was the elastic deformation of the bonds limited by their relatively low strength (*σ^c^* < 0.01*E^b^*), the breakage appeared as clearly brittle (the case of *σ^c^*/*E^b^* = 0.004 in Figure 10b). Therefore, brittle behaviour appears for high bond stiffness and low strength.

### 4.2. Effect of Compaction Pressure

This section of the study concerned the impact of compaction pressure on the tensile strength of the agglomerates. The dependences of the bond cross-sectional area *A^b^* and the bond coordination number *BCN* on the compaction pressure *σ_z_* were considered as the relationships explaining the effect of the compaction pressure on the tensile strength.

#### 4.2.1. Bond Cross-Sectional Area and Bond Coordination Number

The cross-sectional area of the bond (*A^b^*) was assumed to be equal to the average contact area of the particles, and was approximated as
(13)$$A^{b}=\pi \bar{r}{\bar{\delta }}_{n,0},$$
where $\bar{r}$ is the mean radius of the particles and ${\bar{\delta }}_{n,0}$ is the mean residual overlap, i.e., the mean overlap at the instant of bond initiation. The bond coordination number was determined as the ratio of the doubled intact bonds to the number of particles.

Both the cross-sectional area of the bond, *A^b^* (Figure 12a), and the coordination number of the bond, *BCN* (Figure 12b), exhibited a slower-than-linear increase with the increase of the compaction pressure. The dependences were approximated by the power functions:(14)$$\Delta A^{b}\approx \Delta \sigma_{z}^{\alpha }{,}^{}\;{}^{}\alpha =0.794\pm 0.012^{}(R^{2}=0.999),$$
(15)$$\Delta BCN\approx \Delta \sigma_{z}^{\beta }{,}^{}\;{}^{}\beta =0.725\pm 0.005^{}(R^{2}=0.999).$$

The *BCN* was considerably lower than the coordination number during compaction and unloading, and slightly lower than the coordination number for the agglomerates that were removed from the mould and relaxed.

#### 4.2.2. Impact of the Effective Bond Cross-Sectional Area

The simulated relationships fitted the experimental data of *σ*_1_(Δ*L*/*D*) well for the diametral compression of PS agglomerates in the agglomeration-pressure (*σ_z_*) range of 76–153 MPa, for constant values of the bond parameters: *E^b^* = 220 MPa and *σ^c^* = 12 MPa (Figure 13a). In the case of *σ_z_* = 36 MPa, the tensile strength of the bond had to be reduced to 3.6 MPa to fit the experimental data well. A less than three-fold change in the tensile strength of the bond indicates a qualitative change in the binding mechanisms in low and intermediate regions of the strength–pressure relationship, as described by Alderborn [53].

Taking into account the range of the variability of the parameter-fitting characteristics of individual tablets (Figure 9a) for *σ_z_* = 153 MPa (*E^b^* ∈ 200–300 MPa, *σ^c^* ∈ 10–12 MPa), resulting from the variability of their mechanical characteristics, the applied model provided a decent approximation. To analyse the nonlinear dependence of the tensile strength of the agglomerates on the compaction pressure *σ_f_*(*σ_z_*), according to a uniform formula, *σ^c^* was assumed to be 12 MPa for the entire range of *σ_z_* (Figure 13b). The tensile strength *σ_f_* exhibited a faster-than-linear increase with the increase of the compaction pressure *σ_z_*. The rate of this increase followed the faster-than-linear increase of the effective cross-sectional area of the elastic bonds (Δ*A*_e_) with the increase of the compaction pressure:(16)$$\Delta A_{e}^{}=\Delta A^{b}\cdot \Delta BCN=\Delta \sigma_{z}^{\alpha +\beta },$$
where *α* + *β* > 1.

To verify the separate contributions of *A^b^* and *BCN* to the mechanical strength over the entire range of applied values of the compaction pressure, DEM simulations were performed for a series of values of *r_b_* and *BCN* related to *σ_z_* or $\sigma_{z,\max}^{}$, according to the following scheme (Figure 14):
(17)$$case^{}\;1){:}^{}\;{{}_{}}^{}\sigma_{f}^{}(r_{b}(\sigma_{z}),BCN(\sigma_{z})),$$
(18)$$case^{{}^{}}\;2){:}^{}\;\sigma_{f}^{}(r_{b}(\sigma_{z,\max}^{}),BCN(\sigma_{z})),$$
(19)$$case^{}\;3){:}^{}\;\sigma_{f}^{}(r_{b}(\sigma_{z}),BCN(\sigma_{z,\max}^{})).$$

Change in the normalized tensile strength $\Delta \sigma_{f}^{}(\Delta A_{e}^{})/\Delta \sigma_{f}^{}(\Delta A_{e,\max}^{})$ exhibited a linear relationship with change in the normalized effective bond cross-sectional area $\Delta A_{e}^{}/\Delta A_{e,\max}^{}$ (*R*^2^ = 0.998) for all three cases over the entire range of *σ_z_* (Figure 15). The product obtained from cases 2 and 3, which is denoted in Figure 15 as case 4, is as follows:(20)$$case^{}\;4){:}^{}\;case^{}\;2)\cdot case^{}\;3)/(\sigma_{f}^{}{\left(r_{b}(\sigma_{z,\max}^{}),BCN^{}(\sigma_{z,\max}^{})\right)}^{2},$$

It produced almost the same values of the tensile strength as case 1. This agreement illustrates the separate contributions of *A^b^* and *BCN* to the mechanical strength of the bulk materials simulated by the BPM. The faster-than-linear increase of the tensile strength *σ_f_* with the increase of the compaction pressure *σ_z_* reflects the increase of the inter-particle contact area to a critical point, as described by Alderborn [53].

## 5. Discussion

The mechanical breakage behaviour of compacts is determined by micro-bonding mechanisms [3]. Experiments have indicated that brittle breakage is typical for dense structures with a high bond stiffness [23,54]. Ductile breakage has been observed for loose structures with a low bond stiffness [14,15]. Semi-brittle breakage is typical for intermediate compacted structures with an intermediate bond stiffness [11,17]. This classification was clearly reflected by the results of the experimental study of Ge et al. [51], involving the breakage of 3D-printed agglomerates of dense and loose structures with rigid and soft bonds, which elucidated the effects of the packing density, inter-particle bond strength, and elasticity. The results of our study agree with these findings, as well as the findings of Dhanalakshmi and Bhattacharya [18], and indicate that PS particles bonded together in the presence of different binders can exhibit a brittle, semi-brittle, or ductile breakage mode. This highlights the significant effect of binder properties on the tensile strength [55]. Water acts as both a binding agent (in the case of water-soluble compounds) and as a lubricant. The addition of sugar resulted in a less than two-fold increase of the agglomerate strength, owing to the recrystallisation of the solubilised sugar after cooling and the formation of solid bridges. Similarly, protein fibrils may have created binding sites between particles [50]. Gluten proteins—a group of proteins stored with starch in the endosperm of various cereal grains—have unique viscoelastic properties [56]. The addition of gluten with the steaming of the sample reduced the bond stiffness. This manifested as a four-fold decrease of the breakage strength of the agglomerates, as well as the change of the breakage mode from semi-brittle to ductile.

Originally, the BPM was developed to describe the brittle breakage of rocks [25]. Further study allowed the application of the BPM to be extended to other breakage modes. Kozhar et al. [28] proposed the elastic–plastic model and Maxwell viscoelastic model for solid bonds to reproduce the inelastic deformations in titania. The recent study of Ma and Huang [27] showed that application of the displacement-softening model to the bonds allows for modelling of all three modes of agglomerate breakage—brittle, semi-brittle, and ductile—depending on the softening parameter.

The main novelty of our study is the finding that a similar effect to described by Ma and Huang [27] can be obtained using the linear elastic–plastic contact model coupled with the BPM, with a broadened range of the elasticity and the tensile strength of the bonds. A high elasticity and low strength of the bonds, accompanied by a dense structure, resulted in brittle breakage. Low elasticity and high strength of the bonds, together with a loose structure, resulted in ductile breakage. The criterion *E^b^* = *E* separates semi-brittle and brittle breakage. The rate of breakage of the bonds during deformation can be used to distinguish ductile and semi-brittle behaviour. The similar shapes of the experimental and modelled force–displacement relationships, as well as the outlined mechanisms of breakage at the microscale, confirm the applicability of the BPM for simulating the behaviour of agglomerates under compression in a wide range of breakage modes, ranging from brittle to ductile.

The strength of agglomerates increases with the compaction pressure, up to a limiting value [57,58]. The strength limit is related to exhaustion of the increase of the binding mechanisms and the contact area of the plastic deformation of particles when a very high pressure is reached [59]. Alderborn [53] identified three regions of the strength–pressure relationship of powders’ *σ_f_*(*σ_z_*) low-, intermediate-, and high-pressure regions. In the low-pressure region, the pressure is too low for the particles to cohere into a compact. In the intermediate-pressure region, the inter-particle contact area increases up to a critical point. In the high-pressure region, the maximal tablet tensile strength is reached. The results of our study correspond to the low- and intermediate-pressure regions. The experimental results indicated that at the compaction-pressure (*σ_z_*) range of 38–153 MPa, the tensile strength increased linearly with the increase of *σ_z_*. Below this *σ_z_* range, obtainment of a stable agglomerate with adequate strength was impossible. The DEM simulations indicated a faster-than-linear increase of *σ_f_* with the increase of *σ_z_* in both the low- and intermediate-pressure regions, owing to the dependence of two micro-variables—the bond cross-section and bond coordination number—on *σ_z_*.

A comparison of the results of this study with the ones obtained previously by authors from DEM simulations, using the adhesive contact model and the same parameters (with the exception of the contact model) [43], indicates that the source of the largest difference between the outputs was the lack of shear strength in the case of the adhesive contacts. This resulted in a more round force–displacement relationship for the adhesive model compared with the BPM. Disadvantages of the adhesive model include the appearance of secondary contacts during the deformation of the agglomerates. In the case of long-path particle–particle shear displacements, the secondary contacts may slightly increase the effective tensile strength of the agglomerates. The BPM, in which the bonds are formed once, overcomes this inconvenience of the adhesive model. Moreover, the bond stiffness, which is independent of the particle stiffness, considerably broadens the range of applicability for modelling processes in real solids.

## 6. Conclusions

Experiments and DEM simulations of the diametral compression of PS agglomerates were performed. The study provided deep insight into the interactions in an agglomerate at the microscale, elucidating the breakage mechanisms. According to the results, the following conclusions are drawn.

Potato starch agglomerates may exhibit a brittle, semi-brittle, or ductile breakage mode, depending on the applied binder. Starch agglomerates with a moisture content of 17% behaved as semi-brittle materials. The addition of sugar increased the tensile strength of the agglomerates and resulted in the brittle breakage mode. The addition of gluten significantly reduced the tensile strength and resulted in the ductile breakage mode.The BPM, applied together with the linear elastic–plastic contact model, described the brittle, semi-brittle, or ductile breakage mode, depending on the ratio of the strength to the Young’s modulus of the bond *σ^c^*/*E^b^* and the bond-to-contact elasticity ratio *E^b^*/*E*. A low Young’s modulus and high strength of the bond resulted in the ductile breakage mode. A high Young’s modulus of the bond and high compaction resulted in the brittle breakage mode. Intermediate conditions resulted in the semi-brittle breakage mode.The tensile strength of agglomerates determined experimentally increased linearly with the increase of the compaction pressure. The tensile strength determined via DEM modelling exhibited a faster-than-linear increase with the increase of the compaction pressure, which resulted from the faster-than-linear increase of the product of two micro-variables—the bond cross-sectional area *A^b^* and the bond coordination number *BCN*—with the increase of the compaction pressure.The bonded-particle model is promising for DEM simulations of the diametral compression tests of agglomerates.

## Figures and Tables

**Figure 1 materials-13-00932-f001:**
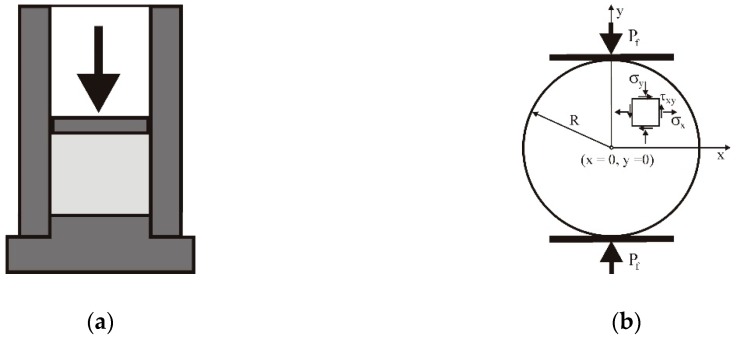
Schematics of the experimental setup: (**a**) compaction, (**b**) diametral compression test.

**Figure 2 materials-13-00932-f002:**
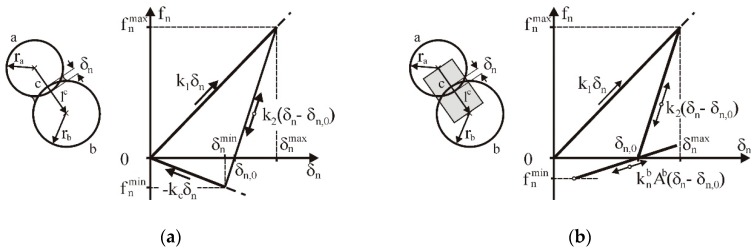
Constitutive linear elastic–plastic contact models: (**a**) linear adhesion model following concept of Luding [35], (**b**) parallel BPM.

**Figure 3 materials-13-00932-f003:**
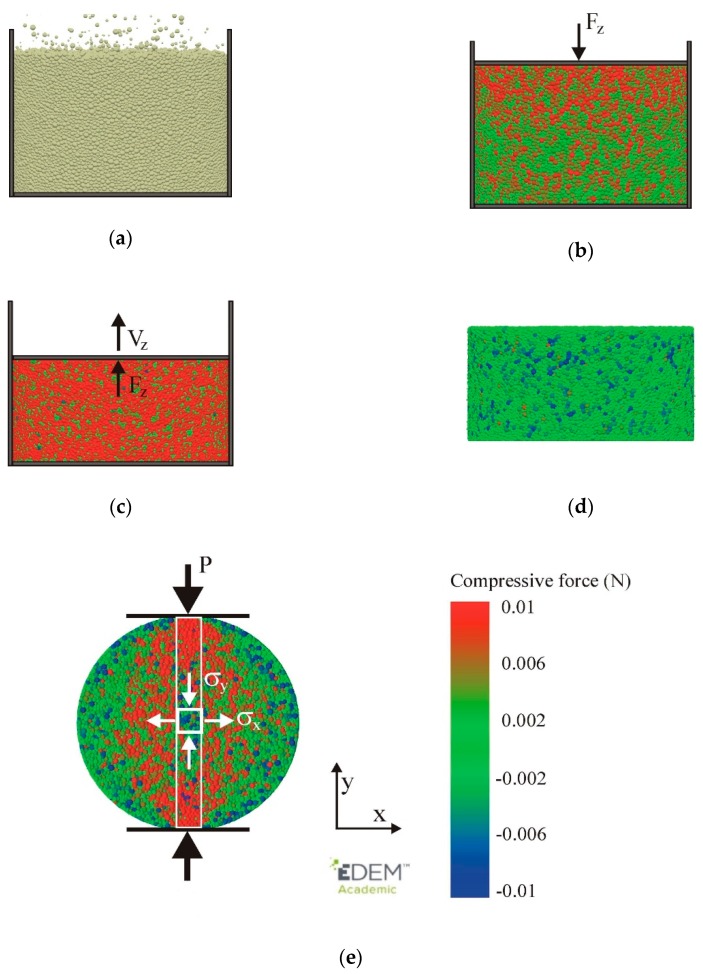
Stages of the simulation: (**a**) filling, (**b**) compaction, (**c**) unloading, (**d**) relaxation, and (**e**) diametral compression modelled with use of EDEM software.

**Figure 4 materials-13-00932-f004:**
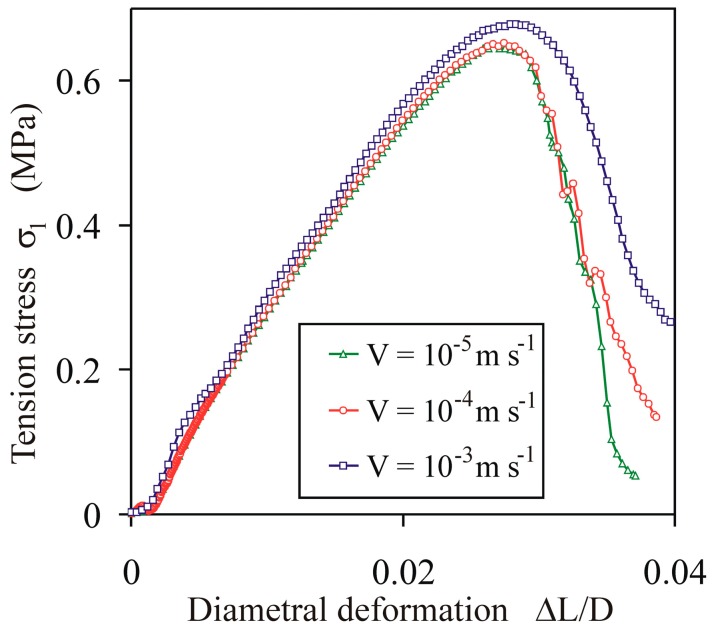
Effect of the displacement velocity *V* on the *σ*_1_(Δ*L*/*D*) characteristic of agglomerates for *σ^c^* = 10 MPa and *E^b^* = 200 MPa.

**Figure 5 materials-13-00932-f005:**
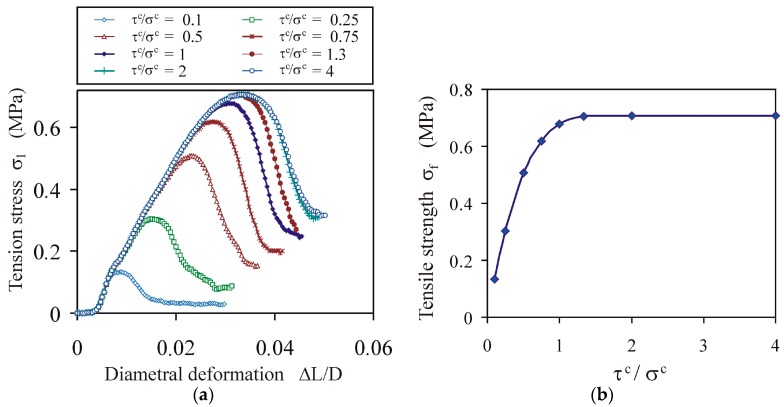
Effect of the *τ^c^*/*σ^c^* ratio on the tensile strength of agglomerates for *σ^c^* = 10 MPa and *E^b^* = 200 MPa: (**a**) tension stress vs. diametral deformation relationships *σ*_1_(Δ*L*/*D*), and (**b**) tensile strength vs. bond shear strength-to-tensile strength relationship *σ_f_*(*τ^c^*/*σ^c^*).

**Figure 6 materials-13-00932-f006:**
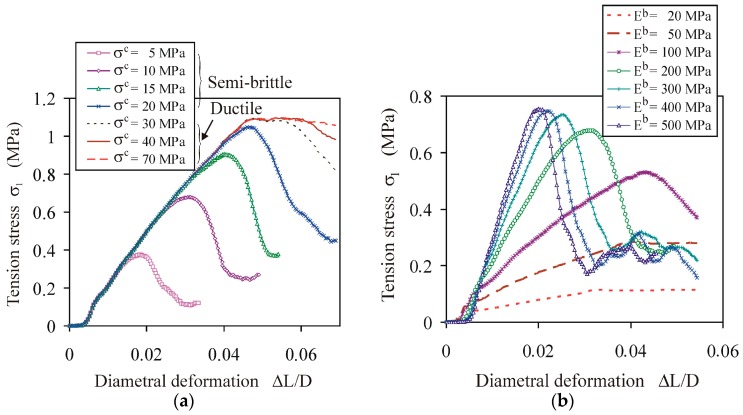
Tension stress vs. diametral deformation relationships *σ*_1_(Δ*L*/*D*): (**a**) effect of the bond strength *σ^c^* = *τ^c^* for *E^b^* = 200 MPa; (**b**) effect of the bond’s Young’s modulus for *σ^c^* = *τ^c^* = 10 MPa.

**Figure 7 materials-13-00932-f007:**
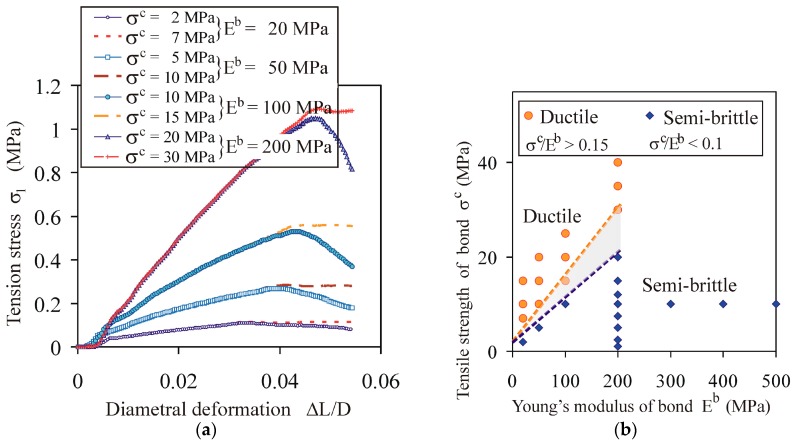
Effect of the *σ^c^*/*E^b^* ratio on the breakage behaviour of the agglomerates for *σ_z_* = 153 MPa: (**a**) comparison of the tension stress vs. diametral deformation *σ*_1_(Δ*L*/*D*) for ductile and semi-brittle breakage of the agglomerates; (**b**) (*E^b^*, *σ^c^*) map of the ductile and semi-brittle behaviour of the agglomerates.

**Figure 8 materials-13-00932-f008:**
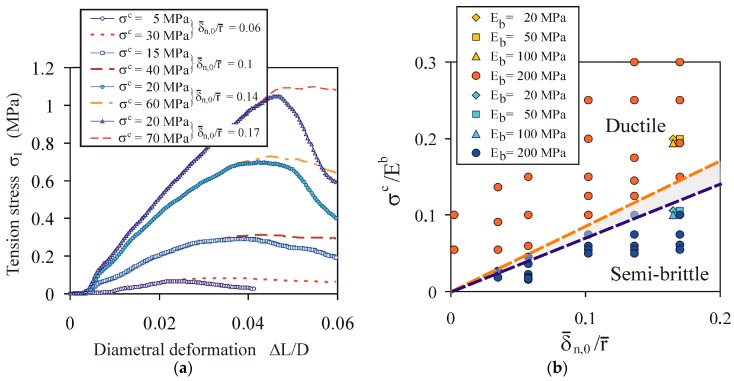
Effect of the *σ^c^*/*E^b^* ratio on the breakage behaviour of the agglomerates during diametral compression: (**a**) comparison of the tension stress vs. diametral deformation *σ*_1_(Δ*L*/*D*) for the ductile and semi-brittle breakage modes at *E^b^* = 200 MPa; (**b**) (*σ^c^*/*E^b^*, ${\bar{\delta }}_{n,0}/\bar{r}$) map of the ductile and semi-brittle behaviour of the agglomerates.

**Figure 9 materials-13-00932-f009:**
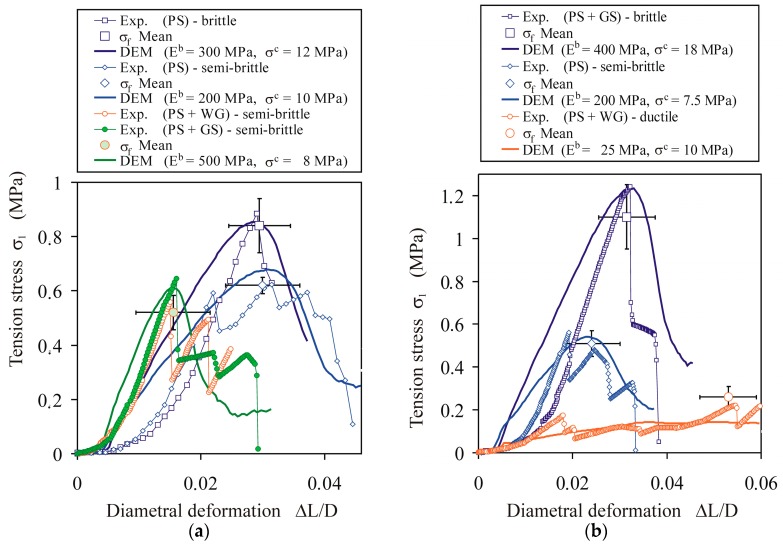
Effects of additives and pretreatment on the *σ*_1_(Δ*L*/*L*) relationships during the diametral compression of the PS agglomerates: (**a**) without steaming and (**b**) with steaming. The bars indicate the standard deviation.

**Figure 10 materials-13-00932-f010:**
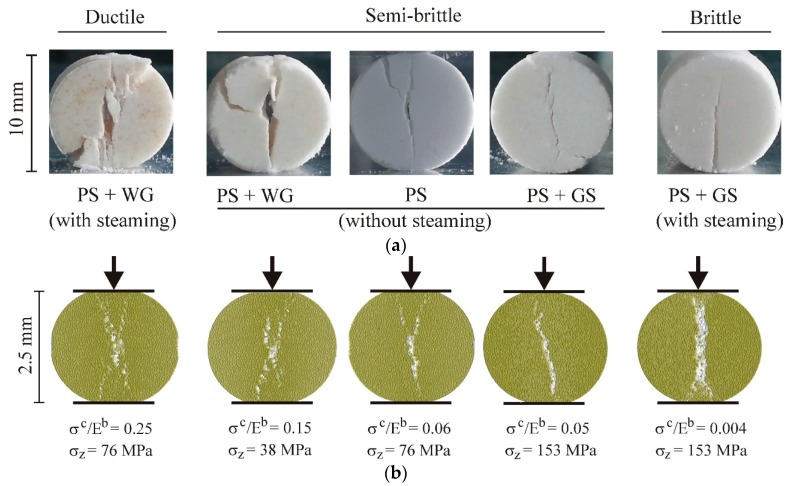
Profiles of the ductile, semi-brittle, and brittle breakage modes: (**a**) experimental and (**b**) simulated for *σ^c^* = 10 MPa.

**Figure 11 materials-13-00932-f011:**
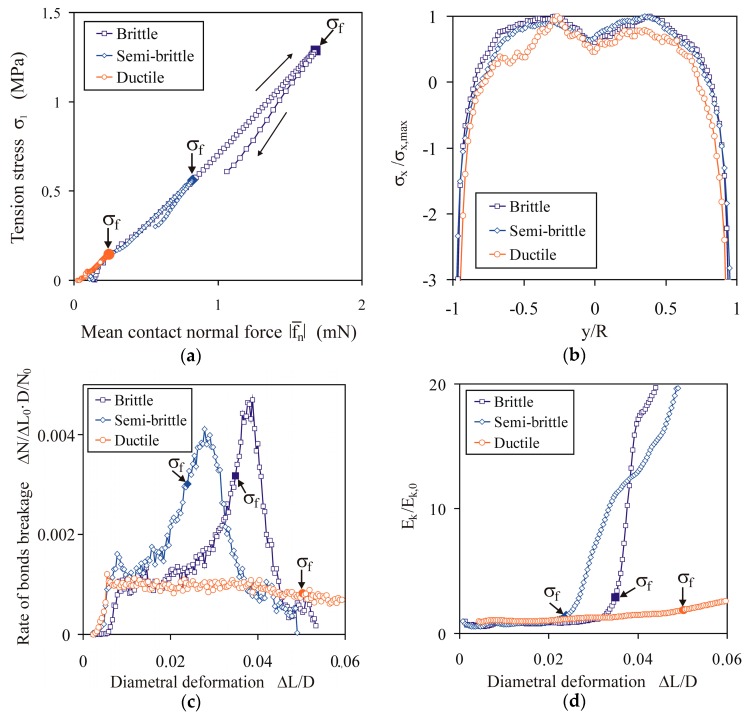
Relationships between micro- and macro-variables for brittle (*σ^c^* = 18 MPa, *E^b^* = 400 MPa), semi-brittle (*σ^c^* = 7.5 MPa, *E^b^* = 200 MPa), and ductile (*σ^c^* = 10 MPa, *E^b^* = 25 MPa) breakage modes: (**a**) dependence of *σ*_1_ on the mean value of the magnitude of the contact normal force $\left|{\bar{f}}_{n}\right|$; (**b**) profiles of the normalized averaged stress component *σ_x_*/*σ_x,_*_max_ along the *y* direction; (**c**) the normalized rate of bond breakages $\Delta N/\Delta L_{0}\cdot D/N_{0}$ vs. diametral deformation, where Δ*N* is the number of bonds broken during the diametral deformation increment Δ*L*_0_/*D* of 3.92 × 10^−5^, and *N*_0_ is the initial number of bonds; (**d**) normalized kinetic energy of particles *E_k_*/*E_k,_*_0_ vs. diametral deformation.

**Figure 12 materials-13-00932-f012:**
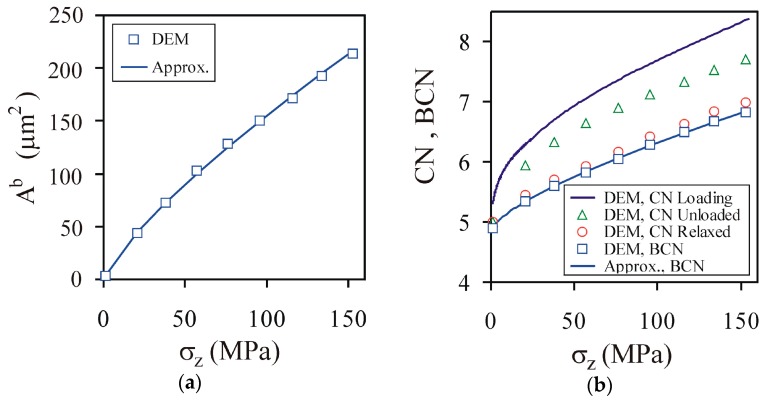
Effects of the compaction pressure on (**a**) the bond cross-sectional area *A^b^*, as well as (**b**) the coordination number *CN* and the bond coordination number *BCN*.

**Figure 13 materials-13-00932-f013:**
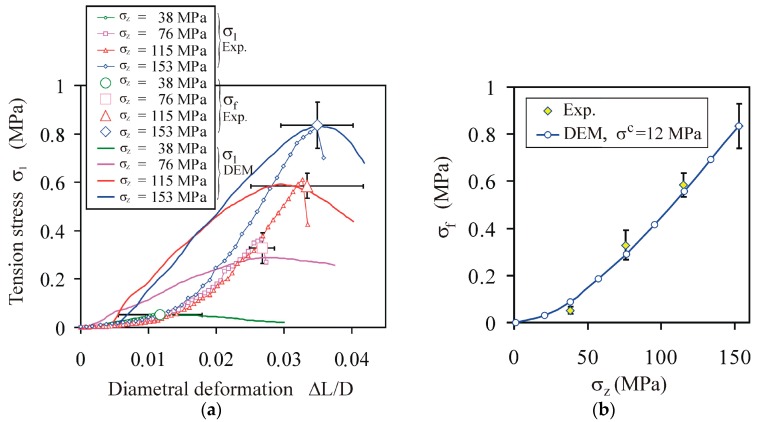
Experimental and DEM-simulated relationships: (**a**) tension stress *σ*_1_ vs. deformation Δ*L*/*D*; (**b**) tensile strength *σ_f_* vs. compaction pressure. The bars indicate the standard deviation.

**Figure 14 materials-13-00932-f014:**
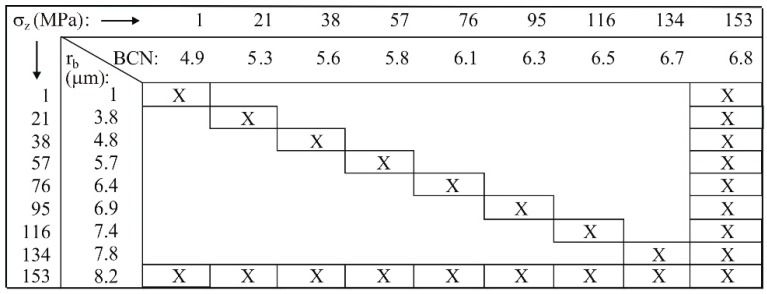
Scheme of simulation variants.

**Figure 15 materials-13-00932-f015:**
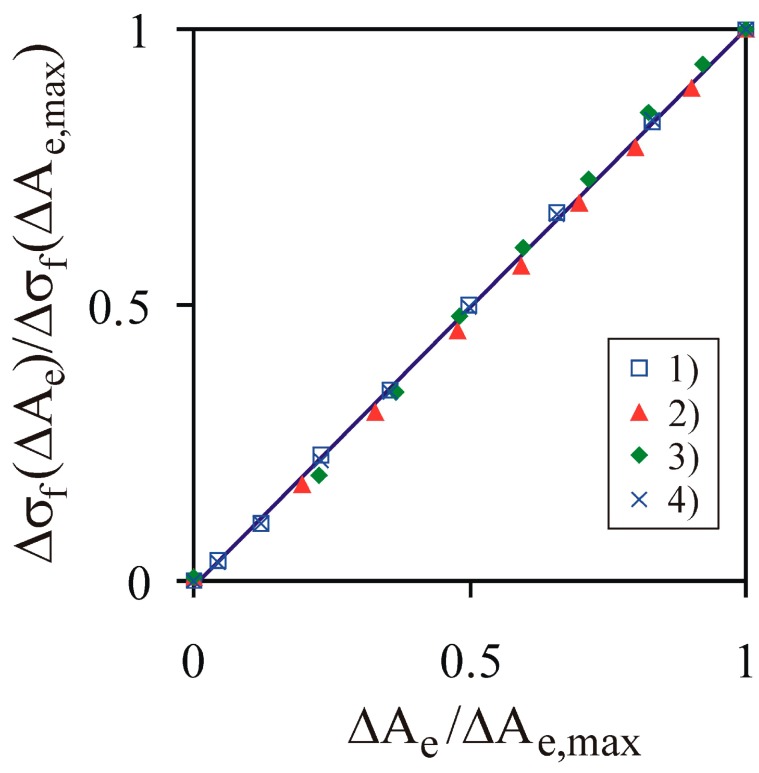
Change of normalized tensile strength $\Delta \sigma_{f}^{}(\Delta A_{e}^{})/\Delta \sigma_{f}^{}(\Delta A_{e,\max}^{})$ vs. change of the normalized effective bond cross-sectional area $\Delta A_{e}^{}/\Delta A_{e,\max}^{}$.

**Table 1 materials-13-00932-t001:** Discrete element method (DEM) simulations parameters.

Parameter	Symbol	Value
Container		
Radius (mm)	*R*	1.25
Height (mm)	*H*	12
Solid density (kg m^−3^)	*ρ*	7800
Young’s modulus (MPa)	*E*	1.561 × 10^6^
Poisson’s ratio	*ν*	0.3
Particles		
Particles number		120,000
Mean particle radius (μm)	*r*	20
Standard deviatio of particle radius (μm)	*r_sd_*	7.5
Particle radius range (μm)		5–36
Particle solid density (kg m^−3^)	*ρ*	1540
Young’s modulus (MPa)	*E*	2.5 × 10^3^
Poisson’s ratio	*ν*	0.25
Yield strength (MPa)	*p_y_*	3 × 10^2^
Mean loading (plastic) stiffness (N m^−1^)	*k* _1_	3 × 10^4^
Mean unloading (elastic) stiffness (N m^−1^)	*k* _2_	1.2 × 10^5^
Mean adhesion stiffness (N m^−1^)	*k_c_*	300
Restitution coefficient	*e*	0.5
Particle–particle friction coefficient	*μ_p-p_*	0.5
Particle–wall friction coefficient	*μ_p-w_*	0.1
Rolling friction coefficient	*m_r_*	0.01
Bond radius (μm)	*r_b_*	1–8.2
Bond tension strength (MPa)	*σ^c^*	2–70
Bond shear strength (MPa)	*τ^c^*	1–40
Bond Young’s modulus (MPa)	*E^b^*	20–2500

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
