# Peer review of "Discrete Element Method Modelling of the Diametral Compression of Starch Agglomerates"

_materials, 2020, doi:10.3390/ma13040932_

Round 1

Reviewer 1 Report

The manuscript entitled "DEM modelling of diametral compression of starch agglomerates" presents results (after both numerical simulations and experimental) revealing the mechanical behaviour of potato starch. The paper is well structured, the discussion is clear and explanatory, the results are new and well presented. In my opinion the manuscript can be published as a scientific paper as it is.

Author Response

The manuscript entitled "DEM modelling of diametral compression of starch agglomerates" presents results (after both numerical simulations and experimental) revealing the mechanical behaviour of potato starch. The paper is well structured, the discussion is clear and explanatory, the results are new and well presented. In my opinion the manuscript can be published as a scientific paper as it is.

Thank you for this opinion.

Reviewer 2 Report

In this manuscript the authors described their investigation regarding the possibility of reproducing different breakage modes of potato starch using discrete element method.  The authors performed a number of simulations mostly relying on the literature and compared the results with the actual experiments they performed on circular specimens. The topic of paper is of interest, the research seems well conducted and the manuscript is properly structured. Besides some minor remarks, I suggest this paper to be accepted and published in the Materials journal.

The abstract is somewhat misleading according to my opinion. It makes the false impression that the paper is mostly about the experiments performed on potato starch, which is also compared to the DEM simulations. In the meanwhile, the paper is mostly about the simulations.

Line 86: If possible, please add the composition of the starch used (amylose/amylopectin ratio, %ash, %lipids, etc.)

Line 98: the authors state that they prepared tablets out of 0.5 g PS. The actual size of the prepared specimens is, however, not clear, only the diameter of 10 mm is specified, the height not. There is a hint about the height being measured with a calliper (which is btw. spelled as caliper in American English), but I found no information later on within the manuscript regarding the height. The 0.5 g of PS used seems also pretty few for a 10 mm diameter, unless we are talking about thin sheets.

Line 237-238: The authors make conclusions regarding the changes in Young’s modulus of the bond, still at the end of the sentence they specify a fixed Young’s modulus of bond and bond strength.

Figure 10a: please insert a scale on the images

Even though the English language of the manuscript is generally good (except some mistakes) the readability is, however, quite poor. The manuscript contains a high number of complex sentences. I faced many parts in the text that I needed to read through multiple times in order to make out their meaning. The presence of the few grammatical mistakes made it even more difficult. A good example for this (line 235-237): The change of the Young’s modulus of the bond for a constant value of the bond tensile strength has an opposite effect to the qualitative change of the breakage behavior of the agglomerates, as the bond strength. (effect on ……than the bond strength),

Author Response

In this manuscript the authors described their investigation regarding the possibility of reproducing different breakage modes of potato starch using discrete element method.  The authors performed a number of simulations mostly relying on the literature and compared the results with the actual experiments they performed on circular specimens. The topic of paper is of interest, the research seems well conducted and the manuscript is properly structured. Besides some minor remarks, I suggest this paper to be accepted and published in the Materials journal.

The abstract is somewhat misleading according to my opinion. It makes the false impression that the paper is mostly about the experiments performed on potato starch, which is also compared to the DEM simulations. In the meanwhile, the paper is mostly about the simulations.

The abstract was modified as follows:

“Starch agglomerates are widely applied in the pharmaceutical, agricultural, and food industries. Formation of potato starch tablets and their diametral compression were simulated numerically and verified in a laboratory experiment to analyse the microscopic mechanisms of the compaction and the origins of their breakage strength. Discrete element method simulations were performed using the EDEM software. Samples comprising 120,000 spherical particles with radii normally distributed in the range of 5–36 mm were compressed in a cylindrical die with a diameter of 2.5 mm. The linear elastic–plastic constitutive contact model with a parallel bonded-particle model (BPM) was used to model the diametral compression. DEM simulations indicated that the BPM together with the linear elastic–plastic contact model could describe the brittle, semi-brittle, or ductile breakage mode depending on the ratio of the strength to Young’s modulus of the bond and the bond-to-contact elasticity ratio. Experiments confirmed findings of DEM simulations and indicated that potato starch agglomerates can behave as a brittle, semi-brittle or ductile material depending on the applied binder. The PS agglomerates without any additives behaved as a semi-brittle material. The addition of 5% of ground sugar resulted in the brittle breakage mode. The addition of 5% of gluten resulted in the ductile breakage mode.”

Line 86: If possible, please add the composition of the starch used (amylose/amylopectin ratio, %ash, %lipids, etc.)

We did not measure the chemical composition of the starch. We found in literature the following data of chemical composition of starch produced from potato cultivated in Poland: amylose/amylopectin ratio 0.266, lipids 0.05%, proteins 0.06%, ash 0.4%, phosphorus 0.08.

Line 98: the authors state that they prepared tablets out of 0.5 g PS. The actual size of the prepared specimens is, however, not clear, only the diameter of 10 mm is specified, the height not. There is a hint about the height being measured with a calliper (which is btw. spelled as caliper in American English), but I found no information later on within the manuscript regarding the height. The 0.5 g of PS used seems also pretty few for a 10 mm diameter, unless we are talking about thin sheets.

Sorry for lack of that information. We corrected the text as follows:

“The height of the tablet after unloading and relaxation, determined by caliper with accuracy of 0.01 mm, was 4.8 ± 0.1 mm.”

Line 237-238: The authors make conclusions regarding the changes in Young’s modulus of the bond, still at the end of the sentence they specify a fixed Young’s modulus of bond and bond strength.

We corrected the text as follows:

“With an increase in Young’s modulus of the bond, the breakage mode changed from ductile to semi-brittle at Eb = 100 MPa.”

Figure 10a: please insert a scale on the images

Thank you for your advice. Figures were corrected.

Even though the English language of the manuscript is generally good (except some mistakes) the readability is, however, quite poor. The manuscript contains a high number of complex sentences. I faced many parts in the text that I needed to read through multiple times in order to make out their meaning. The presence of the few grammatical mistakes made it even more difficult. A good example for this (line 235-237): The change of the Young’s modulus of the bond for a constant value of the bond tensile strength has an opposite effect to the qualitative change of the breakage behavior of the agglomerates, as the bond strength. (effect on ……than the bond strength),

The sentence was corrected as follows:

“The change of Young’s modulus of the bond had an opposite effect on the qualitative change of the breakage behaviour of the agglomerates than the bond strength (Figure 6b).”

We corrected some other spelling mistakes.

Reviewer 3 Report

Overall, this paper describes the application of DEM simulations to the linear-plastic constitutive contact model with parallel bonds (BPM). Previous studies used other models, like adhesive contact model. 

Authors selected well know binders for starch particulates, in order to achieve different mechanical behavior, to have representative data for each behavior and apply the simulations to determine the fitting of the models in real-life formulations. 

The paper is well-written. Is not easy to follow the methodology, but is a topic with a lot of information, considerations and variables. Despite the fact that the reader has to be very focused to understand the thinking process of the authors, the information given is adequate and necessary.

The discussion is well written and the results well exposed and discussed. The withdraw conclusions for this study were well presented to the reader and show that the same methodology can be applied to the different breakage modes, which simplifies the process of simulations depending on the interest of the user of the methodology.

Authors could have tested the applicability of other technologies discussed in the introduction to the experimental data, for comparison purposes. This away would be more evident that the BPM model can be representative in most breakage modes were other models would only respond to specific modes, showing the relevance of the described model. Although these results would improve the quality of the paper, would also bring more complexity, so if intended, the authors could add those results as supplementary material.

Author Response

Overall, this paper describes the application of DEM simulations to the linear-plastic constitutive contact model with parallel bonds (BPM). Previous studies used other models, like adhesive contact model. 

Authors selected well know binders for starch particulates, in order to achieve different mechanical behavior, to have representative data for each behavior and apply the simulations to determine the fitting of the models in real-life formulations. 

The paper is well-written. Is not easy to follow the methodology, but is a topic with a lot of information, considerations and variables. Despite the fact that the reader has to be very focused to understand the thinking process of the authors, the information given is adequate and necessary.

The discussion is well written and the results well exposed and discussed. The withdraw conclusions for this study were well presented to the reader and show that the same methodology can be applied to the different breakage modes, which simplifies the process of simulations depending on the interest of the user of the methodology.

Authors could have tested the applicability of other technologies discussed in the introduction to the experimental data, for comparison purposes. This away would be more evident that the BPM model can be representative in most breakage modes were other models would only respond to specific modes, showing the relevance of the described model. Although these results would improve the quality of the paper, would also bring more complexity, so if intended, the authors could add those results as supplementary material.

Thank you very much for your advice. Indeed, right now we are starting separate project focused on application of BPM to model mechanical strength of agglomerates determined in other tests: uniaxial compression, direct shearing and direct tension. We would like to present this project as a separate publication.